# Ecosystems of Alas Landscapes—The Basis for the Development of Cattle Breeding in the Harsh Natural and Climatic Conditions of the Permafrost Zone

**Roman V. Desyatkin** [1,*] **and Alexey R. Desyatkin** [1,2]

1. Institute for Biological Problems of Cryolithozone SB RAS, 677980 Yakutsk, Russia
2. Melnikov Permafrost Institute of the Siberian Branch of the RAS, 677010 Yakutsk, Russia
* Correspondence: rvdes@ibpc.ysn.ru; Tel.: +7-924-661-4501

**Abstract:** Alas landscapes are unique ecosystems, which are dynamic, geochemically closed thermokarst landforms of the permafrost zone. Alases have a limited capacity in their active layer, and specific conditions for soil, flora and fauna formation. A comprehensive study of alas landscape functions was carried out in Central Yakutia from 1988 to the present time using conventional methods of geobotany, zoology, entomology, etc. This paper presents long-term observations of lake fluctuation cycles and changes in the spatial structure of meadow spaces. The dynamics of the spatial structure lead to significant fluctuations in the productivity of alas phytocenoses. It was revealed that wet and normal alas meadows have the highest vegetation productivity. The long-term course of their productivity tends to decrease, which shows the influence of anthropogenic pressure since the main haymaking areas are located in these meadows. With sharp fluctuations in interannual weather conditions, which determine the microclimatic and soil characteristics of grass growth, the productivity of the edge phytocenoses tends to increase. The productivity of the steppe phytocenoses of the alas remains practically at the same level. Over the years of observation, the economic capacity of alas pastures and hayfields was calculated. Additionally, the paper presents the important role of fauna within closed alas ecosystems, which directly affects the functioning of alas landscapes and is directly involved in soil formation and the circulation of matter and energy.

**Keywords:** Central Yakutia; alas landscapes; thermokarst meadow productivity; alas economical capacity; rodents; entomofauna

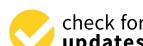



## 1. Introduction

In a zone of permafrost development that has an ice complex, thermokarst landscapes-alases are widespread. The formation of thermokarst landforms and alases began during a warming period between the Pleistocene and Holocene epochs, approximately 8.5–12.8 thousand years ago [1–10]. Currently, in Central Yakutia, the number of alases has reached 16,000, and they occupy 4400 km$^2$, which is 20–30% of the total area [11,12]. Centuries ago, the ancestors of the modern Yakut people tended to the fertile valley and alas meadows of Central Yakutia [5]. On these lands, not only did the formation of the Sakha people take place, but a unique social and economic system was also organized on the basis of animal husbandry in the harsh conditions of the North [13]. Alas meadows are the basis for fodder. Thus, the evolution of permafrost zone with ice complexes, which are the origin of thermokarst depressions, and its significance for the population in the vast territory of the northern hemisphere have been quite fully covered in the recent scientific literature [14–23]. Fluctuations in the exogenous factor parameters (precipitation, duration of the year's warm period, temperature conditions, etc.) cause the interannual dynamics of the lake area, which immediately affect the spatial structure and productivity of meadows [24–30]. The cyclic variability of alas hydroecosystems strongly affects the local land use, which is based on horse and cattle breeding [13]. We estimated that these

alas grasslands can provide enough forage supply for local communities. However, the real alas yield is several times less than the theoretical value because of grassland degradation caused by recent thermokarst processes and waterlogging in the most productive phytocenoses [29]. Therefore, the rapidly changing dynamics of the productivity of meadows in alas landscapes is the main destabilizing factor for the sustainable development of husbandry in taiga-alas landscapes [5]. Additionally, fauna plays an important role in the functioning of alas landscapes (small mammals and insects), as it is directly involved in the soil formation and the circulation of substances and energy [31–34].

Alas landscapes, due to their origin and structural features, are self-organizing, self-regulating and self-developing ecosystems. The main characteristic of alases is the presence of relatively closed, spatially and temporally stable cycles of matter and energy between the biotic and abiotic parts. Alases are characterized by cyclic lake fluctuation, the constant dynamics of the relief, and the migration of periodically appearing and drying lakes. All these factors areconsidered to be a single alas process, which determines theunique features present in soil formation and the functioning of ecosystems in the alas [5,35]. Alas ecosystems are dynamic, often geochemically closed, depressed thermokarst landscapes of the permafrost zone and have a limited capacity in their active layer, which causes specific conditions of soil formation and the formation of flora and fauna [5].

Many scientists, such as geomorphologists, zoologists, geobotanists, etc., have a long-standing interest in how natural processes shape the land surface, hydrology, fauna and flora in alas landscapes. Additionally, they have examined these issues in earlier research, but rarely in an integrated manner. Thus, the value of this work is that all the data presented in this work were obtained from one model alas as a result of comprehensive research conductedover the course ofmore than 30 years.

## 2. Materials and Methods

This study was carried out during 1988–2020 on a typical model alas located 50 km east of Yakutsk on the Tyungyuly terrace of the Lena River. The studied alas is an oval-shaped thermokarst depression, elongated from east to west. The length of the depression space is 240 m, the width is 150 m and the total area of the alas is 11.66 ha. The depth of the depression to the inter-alas plain is 14 m. The alas meadows are mainly used for haymaking and grazing.

The climate of the Central Yakutia is extracontinental, and characterized by severe, dry winters and hot summers [36]. The average annual temperature has been $-9.65\,^{\circ}\mathrm{C}$ for the last 100 years and the annual precipitation has been 235 mm for thelast 50 years (Roshydromet (RIHMI-WDC) database, www.meteo.ru/climate/sp_clim.php first access on 20 August 2021). The territory has a very high annual amplitude of temperature. Therefore, the recorded minimum and maximum temperature is $-63\,^{\circ}\mathrm{C}$ in January and $38.3\,^{\circ}\mathrm{C}$ in July, respectively. The warm period lasts from May to September, reaching the highest temperatures in July (long-term average is $19\,^{\circ}\mathrm{C}$). The coldest month is January with a long-term average of $-39.6\,^{\circ}\mathrm{C}$. About 70% of precipitation falls during the warm period. The maximum monthly average precipitation occurs in July and August, measuring 39 mm for both months. The minimum occurs in February and March, being 8 and 6 mm, respectively [37]. According to Desyatkin et al., 2021, in Central Yakutia during the period of instrumental observations (1930–2022),two shifts in the mean annual air temperature (MAAT) (warming) were marked. The base period from 1930 to 1987 had an MAAT of $-10.3\,^{\circ}\mathrm{C}$. Then, the first shift in the MAAT from 1988 to 2006 presented a sharp increase in the MAAT by $1.7\,^{\circ}\mathrm{C}$ (t-value = $-6.4$, $p < 0.001$). Additionally, the second shift of MAAT in 2007, which lasted till 2022, had an MAAT that amounted to $-7.4\,^{\circ}\mathrm{C}$. A significant statistical difference between these two MAAT shifts (t-value = $-4.7$, $p < 0.001$) was revealed. Thus, over the past 92 years in Central Yakutia, there has been an increase in the MAAT of $2.9\,^{\circ}\mathrm{C}$ [38]. The Kolmogorov–Smirnov Test (K-S test) of normality showed a *p*-value of 0.03010, which provides good evidence that the MAAT data are not normally distributed.

According to the annual precipitation values, there is a slightly positive trend in the last period (2007–2022) due to the precipitation increase in April and October. There was no significant difference in the amount of precipitation between the periods of 1988–2006 and 2007–2022, as the t-value were 0.4 0.3, respectively, with a $p < 0.05$. The value of the K-S test statistic (D) is 0.06063 with a $p$-value of 0.87916, which shows that precipitation data had good normality during the two mentioned periods [38].

Soil properties were studied by digging a full profile, describing the profile and sampling. Carbon and nitrogen content was measured using the C:N analyzer Sumigraph NC-1000 (Sumika Chemical Analysis Service, Osaka, Japan). Soil moisture was measured via the gravimetric method. During the entire period (1988–2020), a field survey was carried out to assess the spatial distribution of different meadows and lake areas inside the thermokarst depression using a theodolite (2T30P, UOMZ, Ekaterinburg, Russia). The measurements were carried out in July, since the difference in vegetation cover among phytocenoses is most clearly observed during this month.

During the field work, geobotanical descriptions were made in accordance with methodological guidelines [39].In conducting botanical research, the method of trial plots was used. In every typical belt of alas meadows, 2 control plots with an area of 5 m$^2$ were laid out annually. Within the plots, grass species were determined using the "Manual of higher plants of Yakutia" [40–42]. To assess the projective cover, the Braun-Blanquet abundance scale was used. The value of the aboveground phytomass of the herbage was estimated by using the method of accounting for plots that were with a size of 1 m$^2$ in 4 replications in the air-dry state [25].

Small mammals were caught in the main biotopes by utilizinga trench with a length of 20 m, a width of 20 cm, and a depth of 15 cm that contained two trapping cones installed within it. In addition, Gero spring traps were also used along the transect to catch small mammals; 25 to 50 traps were installed every 5 m [43]. The trapping results were recalculated per 100 trap-days. The names of the species are given according to the reference guide, "Terrestrial Animals of Russia" [44].

Counting the quantity of insects in the grass cover was carried out by mowing with the entomological net, which was followed by recalculation of the results for 50 strokes. The collection of Orthoptera was carried out with an entomological net for a certain period of time, and the results were recalculated for 1 h [45,46]. The Stanchinsky biocenometer was also used, which helped limit the litter area to 1 m$^2$. Further, after careful disassembling the leaves and examining them from both sides, all the insects collected were placed in the killing jar. In each biocenose, 10 samples were taken and the insects from each sample were placed in a separate killing jar [47,48].

A statistical approach to theKolmogorov–Smirnovtest (K-S test) and a $t$-test were used in the analysis of the obtained data.

## 3. Results and Discussion

The spatial structure of the alas ecosystem is complex and includes the following belt taxa (Figure 1):

(1) The lake;
(2) The belt of coastal aquatic vegetation along the edge of the lake;
(3) The belt of wet meadow on excessively moistened soils;
(4) The belt of normal meadow on normally moistened soils;
(5) The belt of steppe meadow on insufficiently moistened soils;
(6) The belt of mesophytic meadow in the edge spaces along the edge of the forest;
(7) Areas of steppe vegetation on the southern-exposed slopes;
(8) Larch taiga of different types on the slope and under the northern-exposed slope.

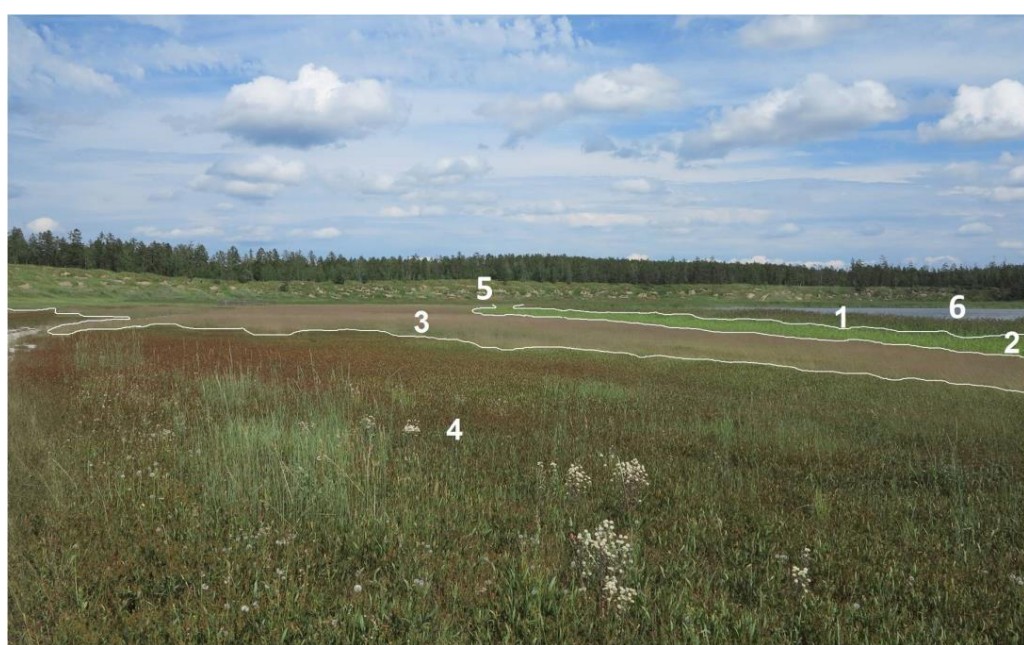

**Figure 1.** Spatial structure of a typical alas in Central Yakutia: 1—belt of coastal aquatic vegetation; 2—belt of wet meadow; 3—belt of normal meadow; 4—belt of steppe meadow; 5—steppes on the southern exposure slope with baidzherakhs; 6—lake.

The degree of individual belt-taxa participation in the formation of the soil and vegetation cover depends on the water supply, the size and configuration of the lakes, and the topography of the bottom of the alas depression. Depending on these factors, the zonality of the belts may be expressed in full, or some belts may be represented only by separate fragments. In small alases with only one lake, the fully round circle belts are more visible. In large alases, the spatial structure of the soil and vegetation cover consists of second-order belts that formed around individual lakes.

The lake water surface area during the observation period of 1988–2020 varied from 0.03 (1988) to 6.89 (2008) hectares, increasing to be more than 200 times higher (Figure 2). Additionally, this increase is combined with the climate shift that begun in2007 and has continued to the present. During this period, increased precipitation has resulted in the alas lake having a stable high level. Depending on the moistening degree of the active layer, the limits of change in the area of wet meadow also reached significant values (up to 14 times). The wet meadow area, which occupied 0.24 hectares in 1988, expanded to 3.36 hectares with the onset of more favorable conditions in 2003. In subsequent years, due to the increase in the lake water surface area, a gradual reduction in the wet meadow area was observed. Due to the rapid expansion of the lake surface area in 2006, this belt disappeared from the alas structure. During the observation period, the minimum limits of area fluctuations were observed in the mesophytic meadow at the forest edge. Its smallest area was equal to 0.61, whereas the maximum was 3.21 hectares, implying that they increased by 5 times. The small fluctuation in the meadow area along the forest edge is explained by its ecological niche, which corresponds to the most optimal conditions for heat and moisture supply compared to other alas phytocenoses. The area of normal meadow changed 13 times, and steppe meadow-7 times.

The rapid dynamics of the alas spatial structure lead to significant fluctuations in the productivity of alas phytocenoses [49]. The phytocenosis species richness within the coastalaquatic belt was very low. The presence of four species of herbaceous plants was noted: *Scolochloa festucacea*, *Scirpus lacustris*, *Bolboschoenus compactus* and *Alopecurus arundinaceus*. The projective cover in different parts of the belt was 55–65%, and the average height of herbaceous vegetation was 50–140 cm. The productivity was 3.0–3.5 T/ha. Hay harvesting is impossible here because these areas are occupied by shallow water; thus, the

grassy cover of alas lakes annually remains and decays to its roots, contributing to the accumulation of peat mass (Figure 3).

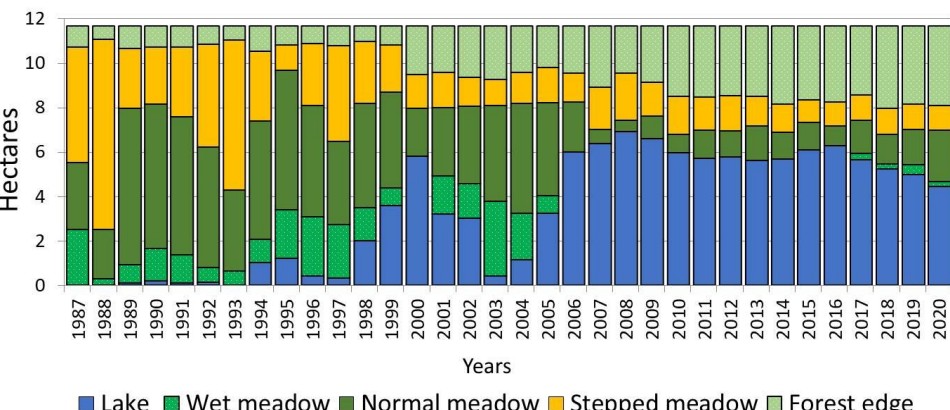

**Figure 2.** Dynamics of the model alas spatial structure.

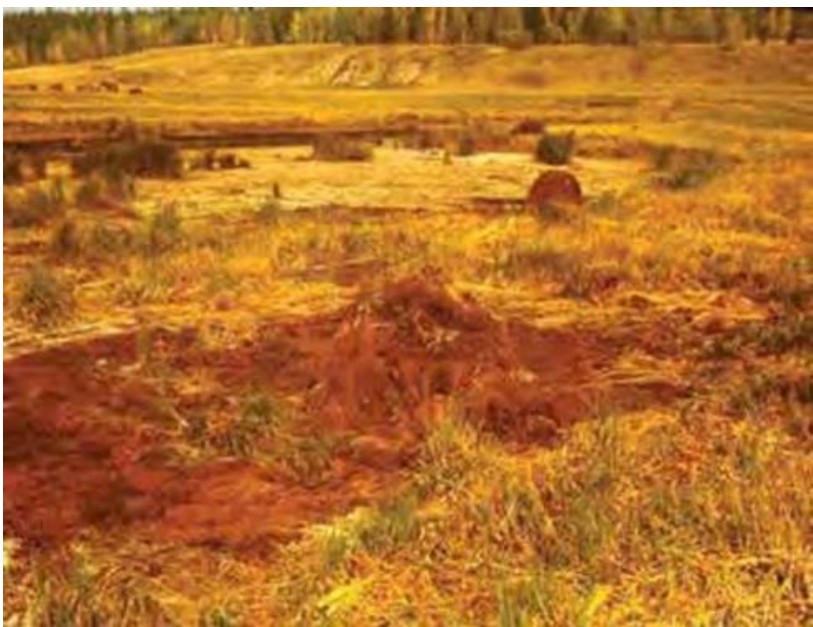

**Figure 3.** Mounds of peat mass on the shore of a drying alas lake.

Wet and normal meadows are the most productive alas phytocenoses. The floristic species richness of wet meadow phytocenoses is much richer than that of the belt of coastal aquatic vegetation; the presence of 28 species of herbs was noted here [25]. The dominant position among cereals is occupied by *Alopecurus arundinaceus*, *Beckmannia syzigachne* and *Poa palustris*. The most characteristic are such representatives of grasses as *Potentilla anserina*, *Polygonum amphibium* and *Inula britannica*. The remaining species can be classified as differential. The projective grass cover is in a range of 60–75%, and the average height was 40–70 cm. The range in alas wet meadow productivity during the observation period varied from 0.43 to 5.02 T/ha for air-dry matter, with an average of 3.11 ± 4.1 T/ha. The minimum productivity values were noted in 1988 and 1993. Although these years were characterized as wet and warm, a massive outbreak of locusts had a negative effect at that time. The C content in this soil amounted to 65.19 kg/m$^2$ at the organogeneous horizon. Carbonate C was measured at 2.25 kg/m$^2$. The amount of N in the soil was 5.39 kg/m$^2$ at the same horizon.

Salt-tolerant species play a dominant role in the formation of phytocenoses within the normal belt of alas meadows [26]. Among the alas cereal plants, the most salt-resistant is *Puccinellia tenuiflora*, which forms continuous thickets. In other cases, this species also forms almost homogeneous communities with the participation of another salt-tolerant species, *Polygonum sibiricum*, in the lower layer of the herbage. In places with the highest degree of soil salinity, a small amountof *Glaux maritima* is typical. The presence of 28 species of grasses was noted in this belt. The projective grass cover was 70–80%, with the average heightbeing 35–45 cm. In favorable years, the normal meadow belt of an alas can produce up to 4.5 or more tons of aboveground biomass per hectare. Thus, the maximum productivity of normal meadow was noted in 2006 and was equal to 4.56 T/ha. In 1988 and 1993 there was a decrease in productivity up to 0.15–0.65 T/ha, which was the same in all alas belts due to a mass outbreak of locusts. Wet and normal meadows have no significant statistical differences in phytocenosis productivity, as the t-value = 2 at $p < 0.05$. The carbon content in the normal meadow soils strongly depends on hay harvest, because this meadow is the main source of forage. Although it has almost the same vegetation productivity as the wet meadow, the carbon content in the soil is twice as low here. The organogeneous horizon contains 30.21 kg/m$^2$ of C and 4 kg/m$^2$ of carbonate C. The N content was 2.39 kg/m$^2$ on the same horizon.

In third place with regard to productivity is the meadow on the edge of forests under the southern-exposed slope [30]. The community present in the mesophytic meadow of marginal spaces was characterized by the highest species richness. The presence of 54–58 species of herbaceous plants was noted here. The composition of phytocenoses of marginal spaces includes facies features. Therefore, in the warmest habitats of the northwestern and northern edges of the alas, the dominant species are *Elytrigia repens*, *Poa pratensis* and *Linaria acutiloba*. In the meadow communities of the edge spaces of the southeastern, eastern and southwestern parts of the alas, the dominant species are *Thalictrum simplex*, *Artemisia tanacetifolia* and *Potentilla stipularis* with the participation of *Carex praecox* and *Poa pratensis*. The total projective grass cover reached 80–90%, and the average height was 45–50 cm. The minimum productivity of the meadow along the edge of the forest was noted in 1988 and amounted to only 0.22 T/ha; the maximum was recorded in 2006, and was 3.17 T/ha. The soil's C content is almost equal to that of the normal meadow and amounted to 35.53 kg/m$^2$.The N content was 3.17 kg/m$^2$ at the organogenic horizon. This meadow is characterized by low carbonate C, which is very similar to the surrounding taiga forest. It is caused by the position on the topography, which provides good drainage.

The floristic species richness of steppe meadow phytocenoses was higher than that of the normal meadow belt [27]. There was a significant increase in species diversity (78 species of herbaceous vegetation). *Elytrigia repens*, *Poa botryoides*, *Carex duriuscula*, *Artemisia commutate* and *Polygonum sibiricum* are dominant. The projective stand cover was 65–70%, and the average height was 30–40 cm. The productivity of the steppe meadow in the most favorable years can reach 2.13 T/ha, whereas in less favorable years it was only 0.44–0.50 T/ha [28]. Normal and steppe meadows have significant statistical differences in phytocenosis productivity (t-value = 6.1 and $p > 0.01$). The steppe meadow and the southern-exposed slope have almost the same conditions, and there was no significant difference in the productivity of vegetation (t-value = 1.9 and $p < 0.05$). This meadow has an elevated position within the alas depression, and is characterized by the lowest soil C content. This meadow often dries up and the C input from vegetation decreases. Thus, the C content here is only 16.26 kg/m$^2$, but the amount of carbonaceous C is more than in other meadows (15.7 kg/m$^2$). The N content is 1.79 kg/m$^2$ at the organogeneous horizon.

During the dry years, the productivity of phytocenoses in alas ecosystems dramatically decreases. Herein, a decrease in biomass compared to the maximum productivity was noted by 5–15 times, for all phytocenoses. When the drought coincided with the mass outbreak of grasshoppers, the decrease in the productivity of the normal meadow in 1988 reached a record value, amounting to only a thirtieth of the maximum productivity.

Sharp changes in phytocenosis productivity led to fluctuations in the yield of alas forage. The total productivity of the studied alas fluctuated over a wide range during the more than 30 years of observations, taking into account the dynamics of the spatial structure and the productivity of phytocenoses in different belts (Figure 4).

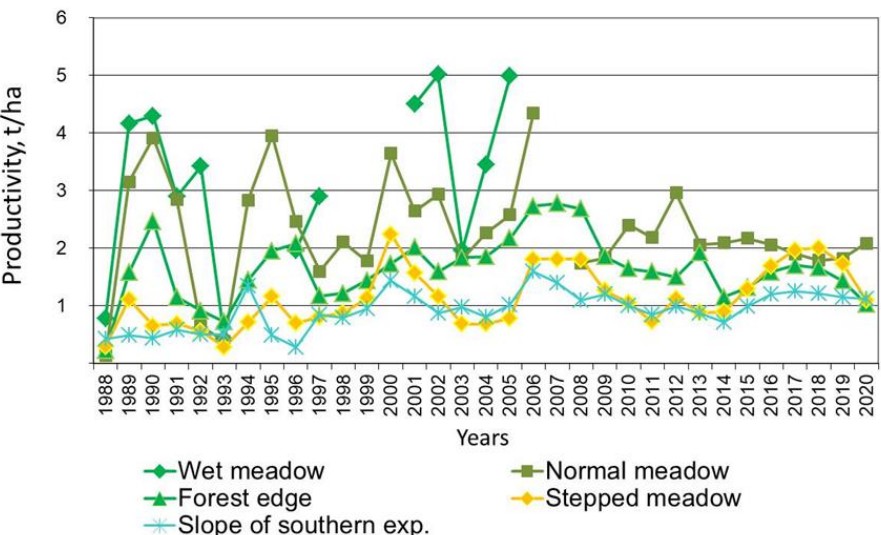

**Figure 4.** Dynamics of phytocenosis productivity of the model alas.

The minimum total yield was measured in 1988 and amounted to 3.44 tons, whereas the maximum in 1990 was 36.22 tons of the 11.66 ha of the studied alas area. The economic (or forage) yield of alas meadows is much lower than the biological productivity, since it depends on the culture of harvesting and tends to decrease. The rapid dynamics of productivity of alas meadows is the main destabilizing factor in the sustainable development of agriculture in taiga-alas landscapes [5]. As a result of long-term observations, it was revealed that the economic capacity of 100 hectares of alas pastures ranged from 14 to 55 heads for cattle in different years, and for horses, it ranged from 10 to 42 heads. The capacity of 100 hectares of alas hayfields for the same period varied from 13 to 65 head of cattle. That is to say, for hay harvesting for the wintering of one conditional livestock head, 2 hectares of hayfields are needed in favorable years and 6–7 in dry years (Figures 5 and 6).

Primary consumers have a significant impact on the productivity of meadows. The rodent fauna in alas ecosystems is quite diverse and is represented by 13 species [50]. Among them, 10 are native and 3 were introduced by humans. The *Ondatra zibethica* was acclimatized for economic purposes. The *Mus musculus* and the *Rattus norvegicus* appeared in settlements as a result of unintentional introduction. The basis of the fauna is made up of widespread species, including the *Microtus gregalis* and the *Microtus oeconomus*. Of the insectivorous animals belonging to the family of shrews, four species live in alas ecosystems: *Sorex caecutiens*, *Sorex daphaenodon*, *Sorex minutissimus* and *Sorex tundrensis*. As a rule, they live on the edges of larch forests, bordering the alases. Shrews are found in all belts, but their population density is quite low. The largest biomass of mouse-like rodents and shrews was observed in the second half of summer and autumn in the wet meadow. Over the years of observations, the biomass varied from 0.46 to 2.69 kg/ha. The minimum biomass of animals was in the belt of the steppe meadow, where only 0.04–0.10 kg/ha was accounted for per hectare. The normal meadow belt occupies an intermediate position, where rodent biomass production was 0.41–1.64 kg/ha. During the study, the maximum population biomass of the *Microtus gregalis* reached 4.62 kg/ha in the alas ecosystem.

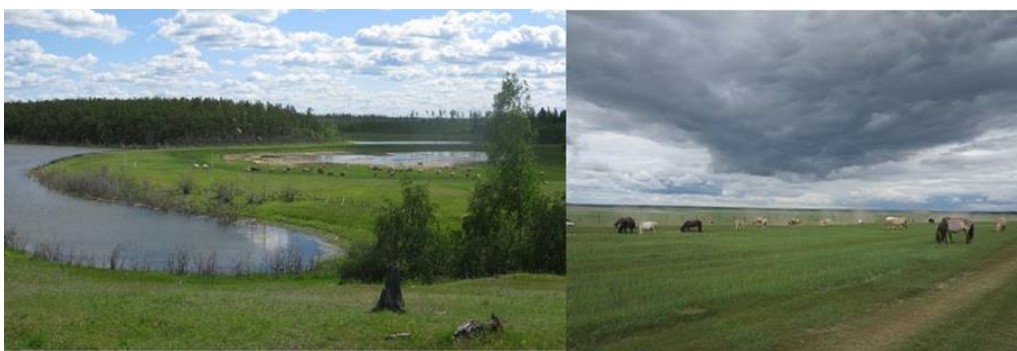

**Figure 5.** Meadows of alas ecosystems are good pastures.

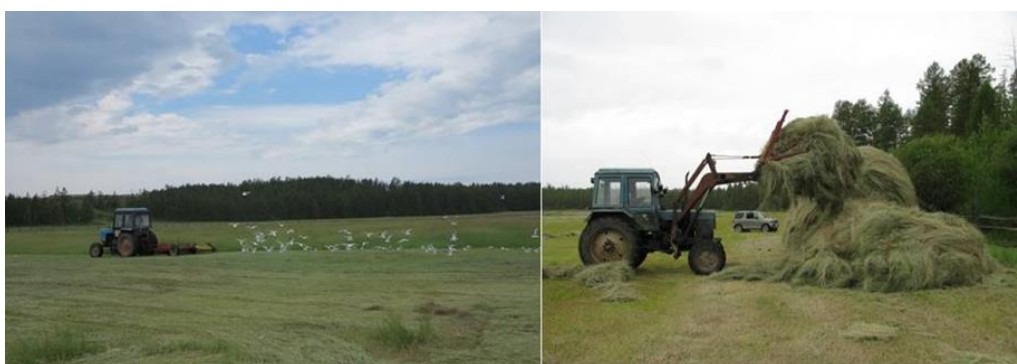

**Figure 6.** Hay harvesting in the alas of Central Yakutia.

The entomofauna can make a serious adjustment to the overall productivity of meadows in alas ecosystems. In the model alas, the presence of 645 species of insects from 389 genera was investigated. There were 105 families, including: Dragonflies (10 species from 3 families and 5 genera), Orthoptera (25 species from 3 families and 13 genera), Homoptera (66 species from 8 families and 47 genera), Hemiptera (132 species from 1 family and 88 genera), Coleoptera (224 species from 39 families and 122 genera), Caddisflies (7 species from 2 families and 4 genera), Lepidoptera (73 species from 18 families and 57 genera), Hymenoptera (25 species from 7 families and 19 genera) and Diptera (71 species from 8 families and 32 genera). The structure of the mesofauna population of alas meadows includes 10 orders of insects and spiders [51]. The most important place among them belongs to Orthoptera, namely, the *Chorthippus albomarginatus*, which is characterized by periodic outbursts of breeding on a catastrophic scale. The seasonal course of the *Chorthippus albomarginatus* population usually has the form of an M-shaped curve, with peaks occurring at the end of June and the second half of July. In some years, the second peak was not traced.

The created long-term database on the quantitative and qualitative characteristics of soils, plants and mesofauna of alases indicates a slow turnover rate and a low volume of the substances cycle in the taiga-alas landscapes. Through the course of natural rhythms in alas soils, there are periods with an acute shortage of mobile forms of mineral and especially organic substances that are necessary for producers for the formation of primary production. Under these conditions, to help the ecosystem return from the critical state, the natural mechanism of regulation was activated in the form of an outbreak of locust mass reproduction [52]. During such periods, their abundance in alas meadows reaches 1500 ind./m$^2$, and their biomass reaches 150 kg/ha (during the normal periods it does not exceed 15–25 kg/ha). An even greater organic matter was found in the green mass of herbs processed by them, which enters the soil in the form of excrement. The biochemical composition of locusts is very favorable: there is 2–2.5 times more fat in grasshoppers than in the dominant grass species in the heading phase (*Puccinellia*, *Alopecurus*, *Beckmannia*,

*Artemisia*, etc.), 2–3 times more crude protein than in herbs, 3 times more digestible protein and 2–2.5 times more total nitrogen. The biochemical composition content in the locust excrement was higher than that of herbs: by fat > 3 times, protein for 1.1 times, crude protein for 1.5–2 times and total nitrogen > 2 times.

A high degree of nitrogenous substances accumulated by insects was noted. Therefore, if the sum of amino acids in hay is 40–45 g/kg, then in locusts it is 300–350 g/kg and in excrement it is 66–70 g/kg. Among these substances, there are many aspartic and glutamic acids, such as serine, threonine, glycine, lysine and arginine. The elemental composition of the locusts and their excrement is also very rich. Thus, in the locusts there was 8.5 times more phosphorus than in hay; 1.5–2 times more potassium; 2 times more vanadium, chromium, cobalt, copper, nickel, lead and molybdenum; and almost 13 times more zinc. The material accumulated in the organisms themselves and their excrement is composed of easily decomposable forms of organomineral substances, which, entering the soil, enrich it with the components of minerals and organic plant nutrition and give impetus to the acceleration of substance circulation.

The after-effects of the locust mass reproduction outbursts had a positive effect in the form of increased productivity during the next 2–3 years, especially in the normal meadow belt. This, apparently, was the result of the enrichment of soil with nutrients that was caused by the excrement and biomass of locusts. Based on this, the ecological role of locusts in the structure and functioning of low-stable, low-productive alas ecosystems of Central Yakutia can be defined as "forming role of mechanism for natural self-regulation and acceleration of the biogeochemical circulation of substances in the extreme conditions of the North".

Natural fluctuations in alas meadow productivity in recent decades have been exacerbated by the negative impacts of humans. The lack of a system for the rational use of biological resources has led to the mass extermination and disappearance of wildlife animals, and the acclimatization of an invasive species, such as *Ondatra zibethica*, which had a detrimental effect on alas ecosystems. This new species completely replaced the native species of fauna, the *Arvicola amphibius*, from the near-water spaces of the alas. The *Arvicola amphibius* had a positive effect, especially in wet meadow ecosystems. Under natural conditions, the number of *Arvicola amphibius* in the meadows at the end of the breeding season reached 30–120 individuals per hectare, amounting to a biomass of up to 12 kg/ha. For each hectare of land, there was a range from 300 to 10,000 burrows [31,53]. When digging burrows, the animals loosened and mixed the soil, and several times increased the aeration, water permeability and intensity of soils moisture [54]. The *Arvicola amphibius* consumes from 65 to 150 g of green fodder per day [31,55], and in the winter time the fodder consumption increases 1.5–2 times [56]. A small proportion of the eaten fodder (about 5%) is converted into secondary products, whereas the rest is returned in the form of undigested residues and excrement. These materials, enriched by organic matter, biogenic macro- and microelements, and biologically active substances, play an important role in the biogeochemical processes of soil systems, i.e., the act as agents of soil formation. By improving the water-physical properties, the aeration and drainage of upper soil layers, and the fertilizing of soils with their excrement, *Arvicola amphibius* ultimately participated in the processes of plant residue decomposition and the formation of the humus profile of soils. The biogeochemical contribution of animals was not limited only by the increase in the fertility of alas soils, but was important for the entire territory of the taiga-alas landscape.

The new species, having displaced the native one, not only eliminated its positive effect, but also had a negative impact on ecosystems. *Ondatra zibethica*, which arranges burrows along the shores of alas lakes, contributes to the development of linear soil erosion in an area up to 25–30 m wide in wet and normal meadow belts. At the same time, soil material drained from eroded meadows accumulates in lakes and contributes to their shallowing. Additionally, when the coastal aquatic vegetation is destroyed by the *Ondatra zibethica*, the physical evaporation from the lakes urface increases. According to M.K.

Gavrilova, in Central Yakutia, 350–400 mm of water evaporates from the open water surface during the summer, exceeding the amount of precipitation by a factor of 2–2.5 [57].

Coastal aquatic vegetation, during biomass formation, consumes 150–200 mm of moisture per season. Therefore, water evaporation is reduced by half in lake areas with well-developed vegetation cover. As a result of the displacement of natural ameliorators, such as *Arvicola amphibius*, all alas soils are now excessively compacted. They quickly lose their productive moisture, which increases the drying of the active layer and accelerates the drying of alas lakes. All this leads to a catastrophic decrease in the productivity of alas ecosystems and slows down the intensity of substances and energy cycles in these closed, unique elements of landscapes. To restore the natural state of alas ecosystems, the complete alienation of the *Ondatra zibethica* and the restoration of the *Arvicola amphibius* population in alases are required.

## 4. Conclusions

Alas soils are unique, which is the result of the functioning of the alas process inside a thermokarst depression. The alas process is a natural and historical process that leads to the formation of negative landforms and the formation of heterogeneous deposits in closed or semi-closed depressions. The microclimatic conditions in an alas contribute to the formation of intrazonal phytocenoses in the zone of middle taiga, and includes meadows of different moisture contents. Over the course of natural fluctuations due to weather conditions, the areas of wet, normal and steppe meadow belts of alases change greatly, which leads to a sharp change in the dynamics in the species diversity and productivity of meadows. As a result of the irrational use of the natural environment in the territory of thermokarst depressions, these taiga-alas landscapes have appeared as centers of anthropogenic degradation in recent decades. Additionally, a sharp decrease in the diversity and productivity of the flora and fauna has been noted. Despite the negative influence of climate change and the anthropogenic impact, meadows of alas landscapes still constitute a significant part of the hay and pasture lands of the republic. For the preservation and development of traditional industries in Yakutia, it is necessary to strengthen the scientific research aimed at protecting and rationally using the biological resources of alas landscapes.

**Author Contributions:** Conceptualization, R.V.D.; methodology, R.V.D.; software, R.V.D. and A.R.D.; validation, R.V.D.; formal analysis, R.V.D. and A.R.D.; investigation, R.V.D. and A.R.D.; writing—original draft preparation, R.V.D.; writing—review and editing, R.V.D. and A.R.D.; visualization, R.V.D. and A.R.D.; funding acquisition, R.V.D. All authors have read and agreed to the published version of the manuscript.

**Funding:** This long-term research was supported by the RFBR grant 19-29-05151; registration number AAAA-A20-120061190009-9; FWRS-2021-0026; and state registration number of EGISU: AAAA-A21-121012190036-6.

**Institutional Review Board Statement:** The animal study protocol was approved by the Institutional Review Board of Institute for biological problems of cryolithozone SB RAS (protocol No 44, date 16.01.2023).

**Informed Consent Statement:** Informed consent was obtained from all subjects involved in the study.

**Data Availability Statement:** The data presented in this study are available on request from the corresponding author.

**Acknowledgments:** We are grateful to the researchers of the Institute for Biological Problems of Cryolithozone SB RAS for working on this long-term project. We thank all geobotanists, zoologists, entomologists and other scientists who participated in this work. And all who mentioned in this paper is agree with this paper.

**Conflicts of Interest:** The authors declare no conflict of interest.

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
