# Peer review of "Ecosystems of Alas Landscapes—The Basis for the Development of Cattle Breeding in the Harsh Natural and Climatic Conditions of the Permafrost Zone"

_land, doi:10.3390/land12020288_

Round 1

Reviewer 1 Report

Dear Authors

Thank you for your submission "Ecosystems of alas landscapes - the basis for the development of cattle breeding in the harsh natural and climatic conditions of the permafrost zone". Current form is just addition of manuscript with no specific novelty and contribution to science. Please find the attachment with comments.

Author Response

Dear reviewers thank you for advice. Please check attached file.

Reviewer 2 Report

1. All over the manuscript, the 'is' tense is being used instead of 'was'. Please correct them appropriately.

2. A diagrammatic presentation of material and method would more appropriately improve the quality of the article and its readability.

3. The climate needs to be categorized into 2 or 3 so as to explain what happens during one climatic typology as it is a long-term experiment.

4. The author has not mentioned anything about what is happening with soil carbon and nitrogen. If they have collected the data sets, they can present an overview of it here.

5. The title shows "development of cattle breeding", but the mention of this part is missing in the manuscript. The authors need to verify the title else they may add some content related to it.

Author Response

Dear Reviewer thank you for kind advice. Please check attached files

Reviewer 3 Report

The information on the variations of the ecosystem studied is accompanied by data from disciplinary studies established for a considerable period of time, this condition allows us to recognize that such changes are extraordinary, both for the speed with which they are happening and for the detailed description Of the agents that have enabled such events, I would recommend to the authors that from these results proposals for sustainable management of the meadows be generated, which favor the improvement in the conditions of grazing and hay production.

Author Response

(The authors gave the same response as above.)

Round 2

Reviewer 1 Report

Dear Authors

Thank you for submitting the revised version. The manuscript is improved than 1st version. 

But still changes are needed. 

INTRODUCTION: This section still need attention. Latest citation needed at least.

Discussion: Add citation here and also discuss results not repeat and remove irrelevant text. 

References: Replace old references with latest.

These comments must resolve that will make the article better. I have attached the file for more comments.

Author Response

Dear reviewers, please see personal reply
